# Study on Tribological Properties and Mechanisms of Different Morphology WS_2_ as Lubricant Additives

**DOI:** 10.3390/ma13071522

**Published:** 2020-03-26

**Authors:** Ningning Hu, Xiuheng Zhang, Xianghui Wang, Na Wu, Songquan Wang

**Affiliations:** 1School of Mechanical & Electrical Engineering, Jiangsu Normal University, Xuzhou 221116, China; huning_ning@163.com (N.H.); 18852172825@163.com (X.Z.); m18362982507@163.com (X.W.); 2School of Mechanical & Electrical Engineering, China University of Mining & Technology, Xuzhou 221116, China

**Keywords:** WS_2_, anti-wear, anti-friction, lubricant additives, tribological properties

## Abstract

In the present work, the relationship curve of the coefficient of friction (COF) with varying loads of different morphology WS_2_ lubricating additives in the friction process at various sliding speeds was studied. On this basis, wear marks and elements on the wear surfaces after friction were analyzed, and then the anti-wear and mechanism effects of WS_2_ of different forms in the lubrication process were discussed. Meanwhile, the Stribeck curve was used to study the lubrication state of the lubricating oil in the friction process. It was revealed that the COF of lubricating oil containing lamellar WS_2_ decreased by 29.35% at optimum condition and the minimum COF was concentrated at around 100 N. The COF of lubricating oil containing spherical WS_2_ decreased by 30.24% and the minimum coefficient was concentrated at 120 N. The extreme pressure property of spherical WS_2_ was better than that of lamellar WS_2_, and the wear resistance of spherical WS_2_ was more stable when the load was over 80 N. The different morphology of WS_2_ additives can play anti-wear and anti-friction roles within a wide range of sliding speeds.

## 1. Introduction

Tungsten disulfide crystal has the advantages of a low coefficient of friction (COF), good heat resistance and oxidation resistance, strong bearing capacity, and compressive strength. It is one of the most popular new functional materials in China and abroad [1]. WS_2_ nanoparticles have a multilayer structure and relatively weak adjacent lamellar interactions (via van der Waals forces), which has attracted widespread attention [2,3,4].

In recent years, researchers have tested the lubricating properties of WS_2_ as a lubricant additive. It was found that WS_2_ could maintain excellent lubrication performance in the range of −273–425 °C, which was significantly better than traditional lubricating oil [5,6,7,8,9]. Nano-WS_2_ as a lubricant additive can significantly improve the anti-wear and anti-friction properties of base oils [10,11,12,13]. Zhang et al. tested the tribological properties of WS_2_ nanomaterials as additives in paraffin oil by Universal Micro-Tribotester (UMT). The experimental results show that the addition of WS_2_ nanomaterials can greatly improve the tribological properties of paraffin oil. When the concentration of WS_2_ nanomaterial additive is 0.5 wt%, the tribological performance of lubricating oil is the best [14]. Aldana et al. studied the tribological behavior of WS_2_ nanoparticles by adding WS_2_ nanoparticles into poly alpha olefin (PAO) base oil containing a zinc dialkyl dithiophosphate (ZDDP) additive, and found that the presence of WS_2_ nanoparticles can improve friction and wear under boundary lubrication conditions, and friction film containing WS_2_ is formed on the friction surface [15].

At the same time, the lubrication performance of WS_2_ is related to the contact force. With the same nano-WS_2_ content, WS_2_ can show good lubrication performance within a certain force range, and the higher the force, the better the lubrication performance. However, once the contact force is exceeded, the effect of WS_2_ as a lubricant addition is greatly weakened [16].

It has been found that the size and crystal structure of WS_2_ particles in the medium have different effects on anti-wear and anti-friction performance [17,18,19,20,21,22,23]. When 90 nm of WS_2_ was added to the lubricating oil at a mass fraction of 0.01%, it showed excellent anti-wear and anti-friction performance, and the tribological performance of layered nanometer tungsten disulfide was better than that of lamellar tungsten disulfide [24].

However, studies on WS_2_ are usually limited to the lubricating effect as a lubricant additive, and few studies have been conducted on the tribological properties of micro-nano lubricant additives in different forms under different loads and speeds. According to the literature, the tribological properties of different forms of micro-nano additives are different under different loads and speeds. For example, the performance of MoS_2_ in different forms as a lubricant additive is quite different. Under the conditions of low loads and low speeds, nano-MoS_2_ can improve the anti-friction and anti-wear performance of base oil, while micro-MoS_2_ can increase the wear. Under high loads and high speeds, micro-MoS_2_ exhibits certain anti-friction and anti-wear ability, while nano-MoS_2_ easily leads to lubrication failure [25].

In this paper, the tribological properties of WS_2_ with different morphology under different loads and speeds were studied, and the surface morphology and element composition of wear scars were also analyzed. At the same time, the anti-wear and anti-friction properties of two kinds of WS_2_ as lubricating additives were discussed, and the mechanism was also investigated.

## 2. Experimental

### 2.1. Preparation of Sample Oil

Two forms of WS_2_, lamellar WS_2_ and spherical WS_2_, were selected as lubricating additives in this experiment. These WS_2_ powders were characterized through X-ray diffraction (XRD) (Bruker company, Germany), Scanning Electron Microscopy (SEM) (FEI company in Hillsboro, OR, United States), and Transmission Electron Microscopy (TEM) (FEI company in Hillsboro, OR, United States) to obtain their physical and chemical properties, such as surface morphology, particle size, and chemical composition. The XRD patterns of two kinds WS_2_ are shown in Figure 1. The SEM diagram of lamellar WS_2_ is shown in Figure 2. The lamellar WS_2_ is a layered structure, its thickness is about 2 μm and the lateral dimension is between 5–10 μm. The TEM diagram of spherical WS_2_ is shown in Figure 3. The average particle size of spherical WS_2_ is 90 nm. In this experiment, low-viscosity PAO6 was selected as the base oil. Lamellar WS_2_ and spherical WS_2_ were added to base oil PAO6 at a concentration of 1.5 wt% to prepare sample oil #1 and #2. The sample oil was dispersed by mechanical stirring and strong ultrasonic compounding for 1 h, and finally the sample oil with WS_2_ evenly dispersed was obtained.

### 2.2. Test Method

The tribological properties of WS_2_ in different forms as lubricant additives under different sliding speeds and loads were tested using a UMT-2 tribometer. In the experiment, the kinematic form was reciprocating linear motion, and the single-stroke of the scratch was set to 15 mm. The contact form of friction pair was ball-on-disc. The upper sample was a steel ball processed from high-carbon chromium bearing steel GCr15 with a diameter of 10 mm, a hardness of HRC58–65, and a surface roughness of Ra ≤ 0.02 μm. The lower sample was also a steel disc processed by high-carbon chromium bearing steel GCr15. The diameter of the sample was 25 mm, the thickness was 10 mm, the hardness was HRC60–63, and the surface roughness was Ra ≤ 0.05 μm. Before the experiment, the samples were cleaned with ultrasonic cleaner for 15 min and dried. Six sliding speeds (10–35 mm/s) and six loads (20–120 N) were set for the tests. The specific set values are shown in Table 1. Each group of experiments lasted for 30 min at room temperature, and the relative humidity was 40%–50%. In addition, drip lubrication was applied in the test, and 3–5 droplets of the lubricant were deposited on the flat before starting the experiment. The friction experiments were repeated three times to ensure reproducibility of the results, and the average value was calculated [26,27].

After the friction experiment, the steel disks were cleaned with acetone for 10 min and dried. The depth and width of the wear scars of the disks were measured using a SV-3000 Ultra-Depth Three-Dimensional Microscope (Santoyo Kanagawa Company, Japan). The wear scars on the steel disks were then analyzed with SEM and an electron-probe micro-analyzer (EPMA) to determine the anti-wear properties of lubricating oil. The anti-wear properties of additives can be determined according to the size of the wear scar, wear depth, and material loss.

## 3. Results and Discussion

### 3.1. Anti-Friction Performance

The average COF curve of the sample oil with different forms of WS_2_ with increased loads at given sliding speeds is shown in Figure 4. Meanwhile, in order to understand the lubrication status of the two sample oils under various experimental conditions, the Stribeck curves were drawn, as shown in Figure 5.

It can be seen from Figure 4 that the COF curve changes in the same trend with the load at each sliding velocity. The average COF of sample oil #1 decreased first and then increased as the load increased. The minimum COF was concentrated at about 100 N, and by calculation, the COF was reduced by a maximum of 29.35%. The average COF of sample oil #2 decreased gradually with the increase of load. In the range of the experimental load, the minimum coefficient was concentrated at 120 N, and by calculation, the COF was reduced by a maximum of 30.24%.

As can be seen from the Stribeck curve in Figure 5, the COF of sample oil #1 decreases first and then increases with the increase of load, which indicates that the whole lubrication state transitions from a fluid lubrication state to a mixed lubrication state. This occurs because when the load is small, the thickness of the oil film is thick, and the contact surface of the friction pair is completely separated by the oil film. At this time, the friction force comes from the viscous shear of the lubricating oil. As the load increases, the oil film thickness decreases and the viscous shear force of the lubricating oil also decreases, and so the COF decreases. When the oil film cannot completely separate the surface of the friction pair, the surfaces contact and rub with each other, resulting in an increase in the COF. At this point, the lubrication state changes to the mixed lubrication state. Under the same experimental conditions, the Stribeck curve of sample oil #2 shows that the COF decreases with the increase of load, and the whole lubrication state is basically in a state of fluid lubrication. Because the surface of the friction pair is effectively separated by the oil film in the range of the experimental load, the COF becomes smaller with the increase of the load.

In addition to the change in the lubrication state caused by the thinning of the oil film, it may also be because the frictional heat inside the lubricant causes the bearing capacity of the lubrication film not to increase infinitely. When it increases to the maximum, the oil film will break. However, the lubrication state of sample oil #2 did not change significantly within the same load range. This indicates that the bearing capacity of sample oil containing spherical WS_2_ particles is better than that containing lamellar WS_2_ in this load range.

In conclusion, within the range of experimental load, the lubrication state of lamellar WS_2_ transitioned from a fluid lubrication state to a mixed lubrication state, while the lubrication state of spherical WS_2_ remained basically in a fluid lubrication state and did not changed. According to the experimental result, it can be found that both forms of WS_2_ as additives can affect the anti-friction performance and bearing capacity of lubricating oil, and the spherical WS_2_ shows more outstanding anti-friction performance and bearing capacity.

### 3.2. Abrasion Resistance

The three-dimensional morphology and wear trace diagram of wear scars on steel disks under four different loads (20, 60, 100, and 120 N) are shown in Figure 6 and Figure 7, and the wear rate histogram under different loads is shown in Figure 8.

From Figure 6 and Figure 7, it can be seen that with the increase of the load, the average depth and width of the wear track continuously expand, so it can be concluded that the wear volume of the wear scar also increases continuously with the increase of the load. Combined with Figure 8, the wear rates of wear scars corresponding to the two sample oils tend to decrease first and then increase with the increase of the load. The minimum wear rates were concentrated at a load of about 80 N. The minimum wear rate of sample oil #1 was 10.48 × 10^−4^ mm^3^⋅N^−1^ m^−1^, and the minimum wear rate of sample oil #2 was 10.9 × 10^−4^ mm^3^⋅N^−1^ m^−1^, indicating that the abrasion resistance of the two kinds WS_2_ was similar at this time. However, when it exceeded 80 N, the wear rate of sample oil #2 increased slowly compared with that of sample oil #1, which indicates that the abrasion resistance of spherical WS_2_ was more stable under large loads.

In order to understand the wear form of the wear scars on the surface of the steel ball, the wear scars were characterized. Figure 9 and Figure 10 are SEM-micrographs of the two sample oils under different loads at the same sliding speed. As can be seen from Figure 9 and Figure 10, the main wear forms of wear scars are plastic deformation, furrow cutting, and adhesive wear [28,29,30]. There were obvious furrows on the worn surface, and some of the surfaces were broken. The furrow on the surface of the wear scar expanded with the increase of the load, and the wear became more and more serious, which indicates that the load had a positive correlation with the wear amount. Combined with Figure 8, with the increase of the load, the wear rate first decreases and then increases, indicating that the addition of different forms of WS_2_ can improve the anti-wear performance of lubricating oil within a certain range of the load. When the load exceeded a certain value, the anti-wear performance was limited and the anti-wear ability was weakened.

### 3.3. Lubrication Mechanism

The composition and distribution of elements on the surface of wear marks were analyzed using EPMA to grasp the chemical changes on the worn surface. A wear scar of sample oil #1 with a 100 N load and 15 mm/s sliding speed, and a wear scar of sample oil #2 with a 120 N load and 20 mm/s sliding speed were used as examples. The results of their characterization are shown in Figure 11 and Figure 12. (a)–(e) correspond to the SEM diagram of the wear scar (a) and the distribution diagram of Fe (b), O (c), W (d), and S (e), respectively.

It can be found from the distribution diagram of each element in Figure 11 that elements Fe and O are very densely distributed, the element W is evenly distributed, and the element S is the least distributed. According to literature [31,32,33,34,35], it was found that oxidation products were formed on the friction surface through the X-ray photoelectron spectroscopy (XPS) analysis of the friction surface after the test. After further study on the products formed, it was found that the Fe 2p peak corresponds to Fe_2_O_3_ or FeO, and the worn surface appeared on FeO and Fe_2_O_3_ film. The XPS data indicate that a frictional chemical reaction occurs on the surface of the wear steel during the friction process, thus forming the boundary lubrication film composed of WO_3_, Fe_2_O_3_, and FeO. Therefore, it can be speculated that the friction surface is most likely to form a dense chemical reaction film containing Fe_2_O_3_ and FeO through chemical reactions [31,32,33,34,35]. In addition, the element W is evenly distributed on the friction surface, while the element S is scarce on the friction surface, indicating that W is most likely to form tungsten oxide on the surface. The presence of trace element S indicates that there is still trace lamellar WS_2_ on the friction surface.

From the distribution diagram of each element in Figure 12, it can be found that the distribution of elements Fe, O, and W is similar to that in Figure 11, and the content of element S is higher than that in Figure 11, which is mainly scattered on both sides of the wear scar. It can be inferred that a layer of chemical reaction film containing Fe_2_O_3_, FeO, and WO_3_ is formed on the friction surface. In addition, the distribution diagram of elements W and S indicates that some nano-WS_2_ remained in the grooves on both sides and cracks of the wear scars.

It can be inferred from the elemental analysis of the wear surface that during the lubrication process, the sample oil is likely to form a dense layer of chemical reaction film containing Fe_2_O_3_, FeO, and WO_3_ on the surface. WS_2_ will be deposited on the friction surface to form a physical adsorption film filled in time if there is damage on the surface. Under the combined action of physical deposition film and chemical reaction film, the anti-wear and anti-friction properties of the sample oil were improved. In addition, the residue amount of spherical WS_2_ was larger than that of lamellar WS_2_, which may be because the small size of WS_2_ makes it more prone to deposit in furrows and grooves, and timely fill and repair when surface damage occurs.

The lubrication mechanism of lamellar WS_2_ as an additive is shown in Figure 13. The lamellar WS_2_ dispersed in the lubricating oil is partially deposited on the surface of the friction pair to form a physical deposition film under the action of frictional shear and normal load. Due to the crystal structure of lamellar WS_2_, the layer to layer is connected by weak van der Waals force, and the surface is easy to slip, which leads to the reduction of COF. Moreover, the deposition film prevents the direct contact of the friction surface; therefore, the wear is reduced, and the anti-wear performance of the lubricating oil is improved.

For spherical WS_2_ particles, the lubricating mechanism is roughly shown in Figure 14 when it is used as a lubricant additive. Nanoparticles have excellent adsorption and dispersal: therefore, the spherical nano-WS_2_ dispersed in the lubricating oil will slowly be deposited on the surface of the friction pair to form a certain thickness of physical deposited film during the process of friction, preventing direct contact of the friction pair surface. The spherical structure of WS_2_ can be regarded as a ‘small ball’, which can play a similar role as a ‘micro-bearing’ on the friction surface, changing the sliding friction between the surfaces into rolling friction, thereby reducing the COF. With the action of frictional shear force and normal load, the spherical WS_2_ will be deformed by squeezing and even peeling off. The exfoliated WS_2_ molecular layer will remain on the pits and peak tips of the friction surface, covering the contact point, thereby reducing the direct contact of friction pair, and then reducing friction and wear. In addition, the peaks of some micro-convex bodies on the surface of the friction pair will penetrate the lubricating oil film and directly contact, collide, and break, accompanied by abrasive particles and new wear scars. With the generation of new wear scars, spherical WS_2_ with small sizes will fill in the scratches generated by grinding, playing a role in repairing the friction surface and making the surface smooth: thus, reducing the COF and reducing wear.

## 4. Conclusions

In this paper, the tribological properties of WS_2_ with different forms as lubricant additives under different loads were studied, and the anti-friction performance and lubrication states of sample oil were analyzed through the COF curve and the Stribeck curve. The anti-wear performance of the sample oil was analyzed by the surface morphology and wear form of the wear scars, and the anti-wear and anti-friction mechanism of WS_2_ with different forms in the lubrication process was discussed. The conclusions are as follows:(1)The average COF of lubricating oil containing lamellar WS_2_ decreased first and then increased with the increase of load, and the COF was reduced by a maximum of 29.35%, and the minimum COF was concentrated around 100 N. The average COF of lubricating oil containing spherical WS_2_ gradually decreased with the increase of load, and the COF was reduced by a maximum of 30.24%. In the range of the experimental load, the minimum COF was concentrated around 100 N. In addition, the extreme pressure performance of the spherical WS_2_ was superior to that of the lamellar WS_2_.(2)According to the Stribeck curve, within the range of the experimental load, the lubrication state of the lamellar WS_2_ transitioned from a fluid lubrication state to a mixed lubrication state, while the lubrication state of the spherical WS_2_ basically remained in a fluid lubrication state without any change.(3)With the increase of the load, the average depth and width of the wear track were continuously expanding, and the wear rate of the two sample oils tended to decrease first and then increase with the increase of the load. The minimum wear rate occurred when the load was around 80 N, and the two values were close, indicating that the anti-wear performance of the sample oil was similar at this time. However, when it exceeded 80 N, the wear resistance of spherical nano-WS_2_ was more stable.(4)The main difference in the anti-wear and anti-friction mechanism of WS_2_ with different forms was that, for lamellar WS_2_, its lamellar structure made it easy to slip between layers and the surfaces of layers, leading to the reduction of COF. For spherical WS_2_, its spherical structure can be regarded as a ‘small ball’, which can play a role similar to ‘micro-bearing’ on the friction surface, changing the sliding friction between surfaces into rolling friction: thus, reducing the COF.

## Figures and Tables

**Figure 1 materials-13-01522-f001:**
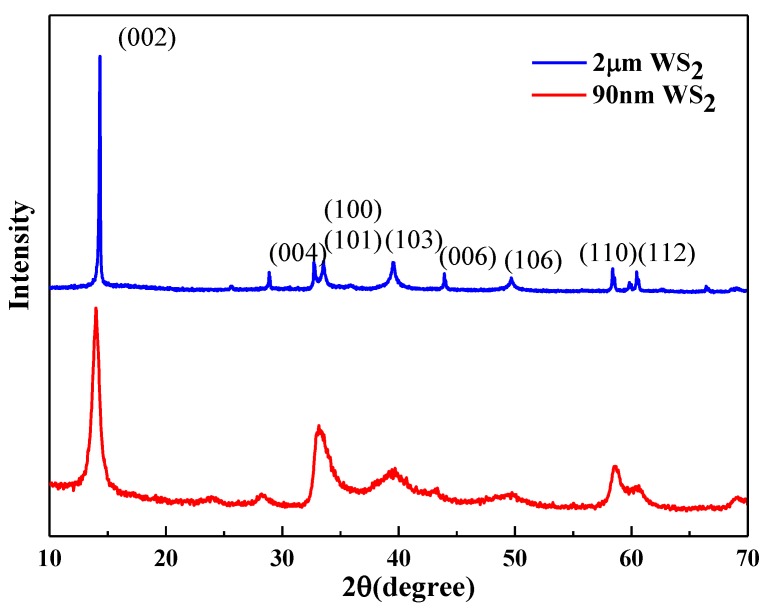
X-ray diffraction (XRD) image of different forms WS_2_.

**Figure 2 materials-13-01522-f002:**
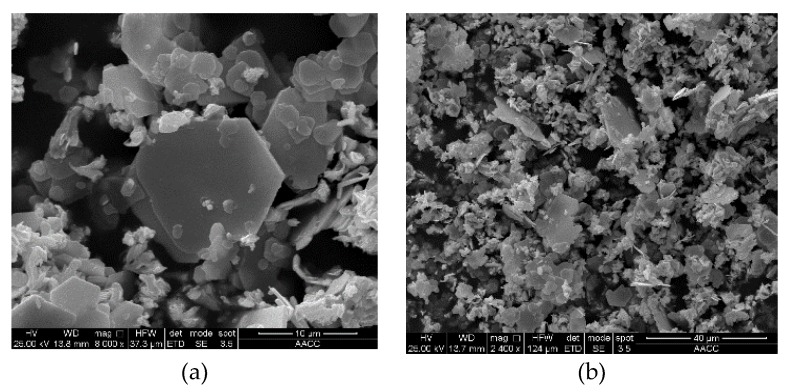
Scanning electron microscopy (SEM) of micro-WS_2_ under different magnifications: (**a**) 8000 ×; (**b**) 2400 ×.

**Figure 3 materials-13-01522-f003:**
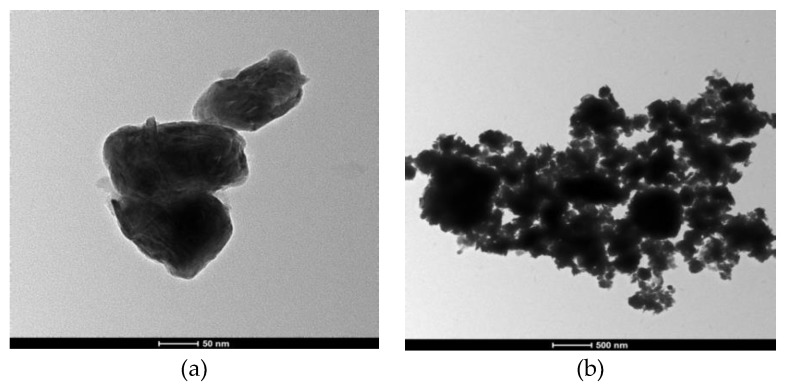
Transmission Electron Microscopy (TEM) with different magnifications of nano-WS_2__._ (**a**) 50 nm; (**b**) 500 nm.

**Figure 4 materials-13-01522-f004:**
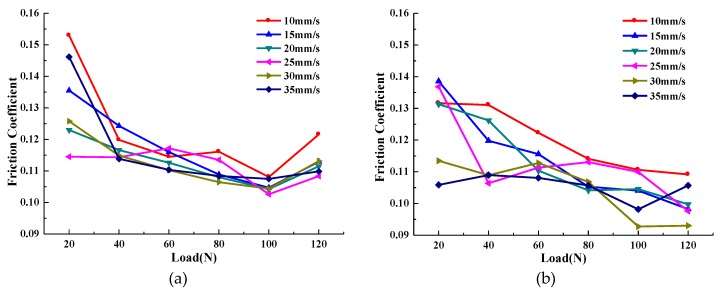
Average coefficient of friction (COF) curves of sample oil varies with loads: (**a**) flake micron WS_2_; (**b**) spherical nano-WS_2_.

**Figure 5 materials-13-01522-f005:**
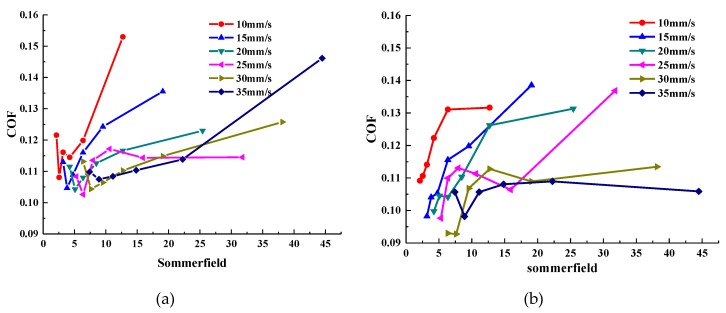
Comparison of Stribeck curves of sample oil at different sliding speeds: (**a**) sample oil #1; (**b**) sample oil #2.

**Figure 6 materials-13-01522-f006:**
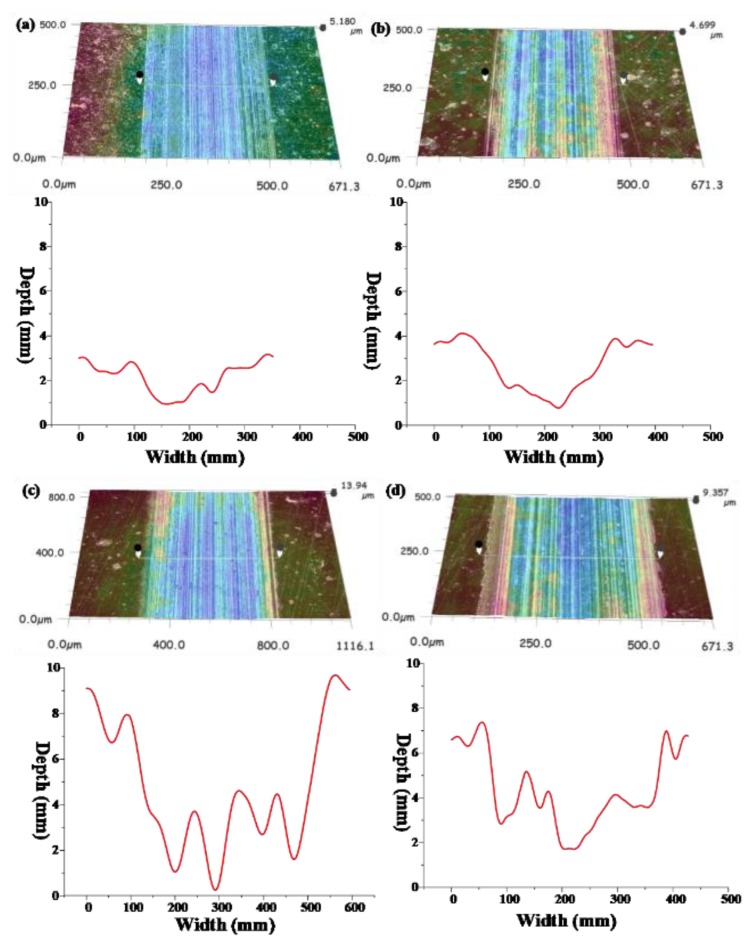
Three-dimensional morphologies and wear tracks of the wear scars corresponding to sample oil #1 under four different loads at the same sliding velocity: (**a**) 20; (**b**) 60; (**c**) 100; (**d**) 120 N.

**Figure 7 materials-13-01522-f007:**
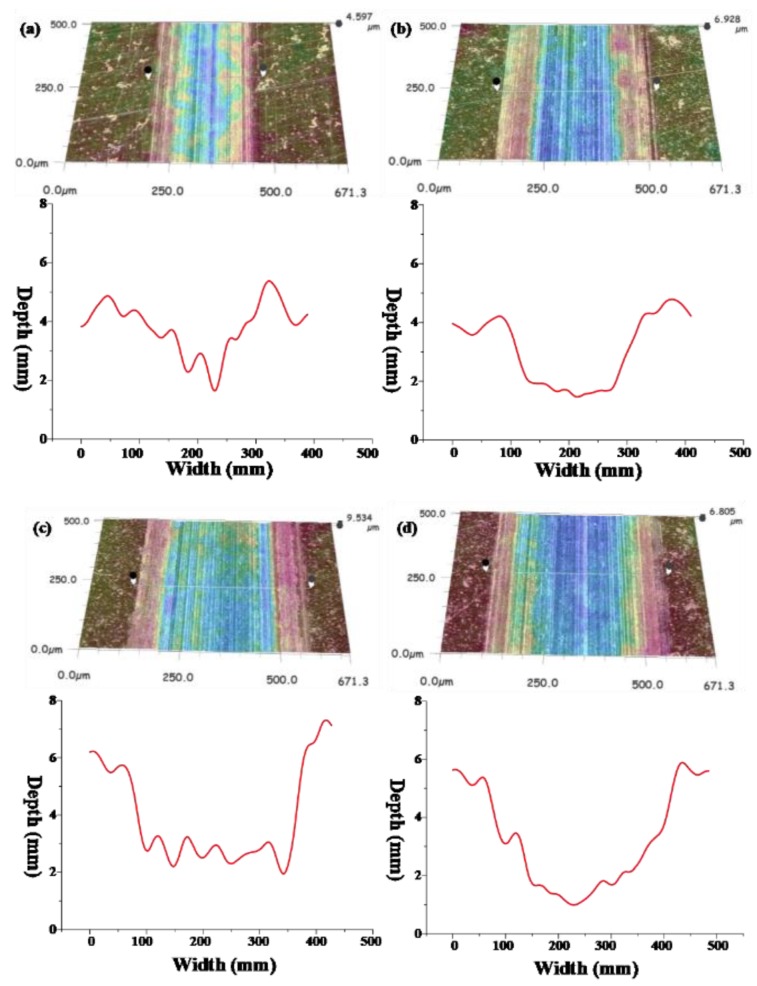
Three-dimensional morphologies and wear tracks of the wear scars corresponding to sample oil #2 under four different loads at the same sliding velocity: (**a**) 20; (**b**) 60; (**c**) 100; (**d**) 120 N.

**Figure 8 materials-13-01522-f008:**
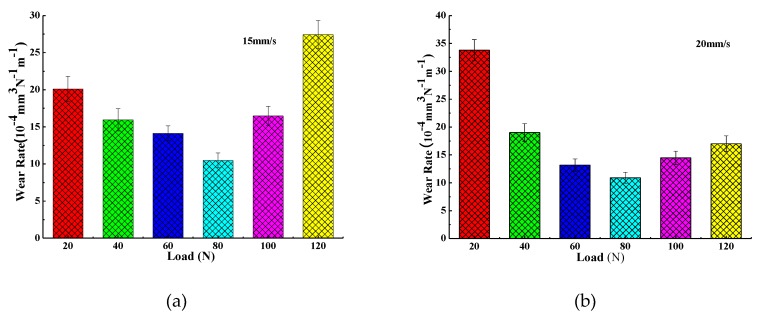
Wear rate histogram of sample oils under different loads. (**a**) sample oil #1; (**b**) sample oil #2.

**Figure 9 materials-13-01522-f009:**
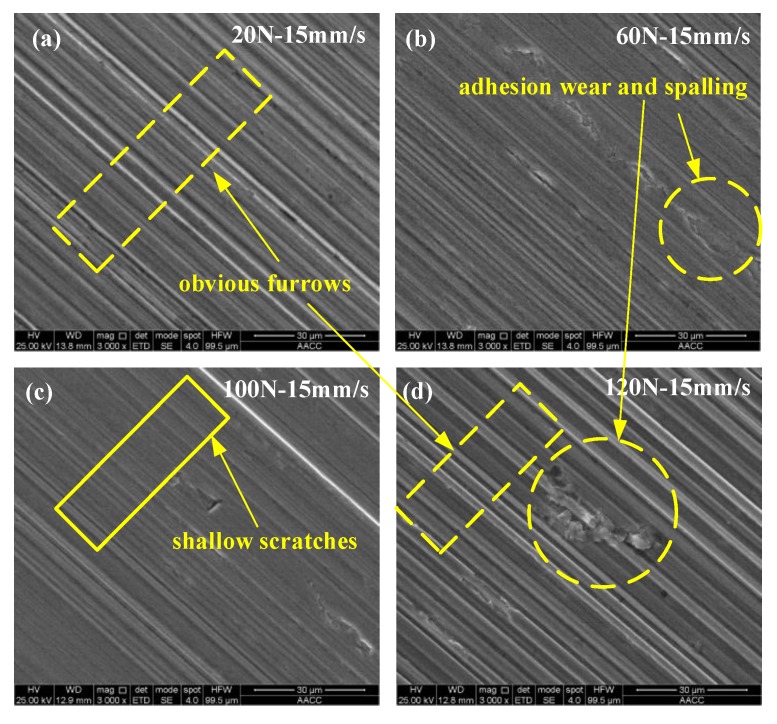
SEM of wear scars of sample oil #1 under different loads at the same sliding speed. (**a**) 20 N-15 mm/s; (**b**) 60 N-15 mm/s; (**c**) 100 N-15 mm/s; (**d**) 120 N-15 mm/s.

**Figure 10 materials-13-01522-f010:**
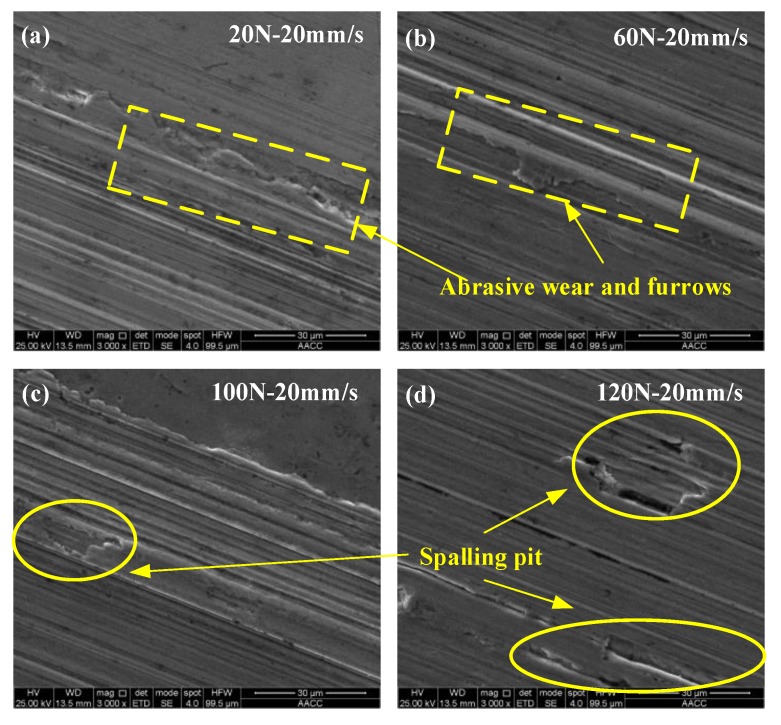
SEM of wear scars of sample oil #2 under different loads at the same sliding speed. (**a**) 20 N-20 mm/s; (**b**) 60 N-20 mm/s; (**c**) 100 N-20 mm/s; (**d**) 120 N-20 mm/s.

**Figure 11 materials-13-01522-f011:**
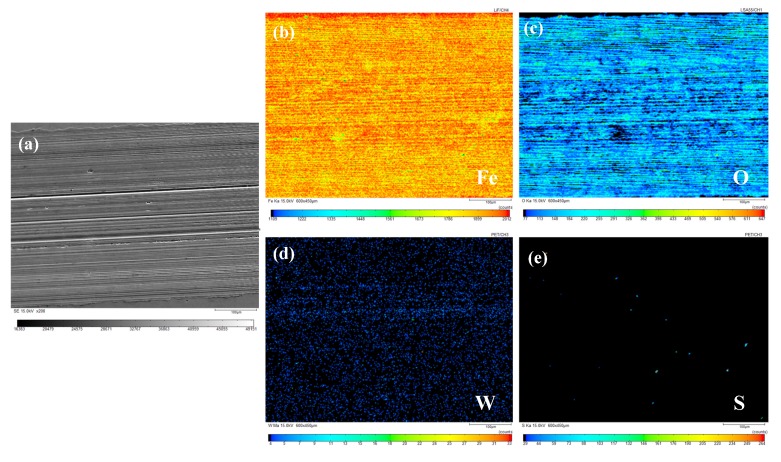
Element distribution diagram of wear scar corresponding to sample oil #1 (load: 100 N, sliding speed: 15 mm/s). (**a**) Scanning image; (**b**) Element Fe; (**c**) Element O; (**d**) Element W; (**e**) Element S.

**Figure 12 materials-13-01522-f012:**
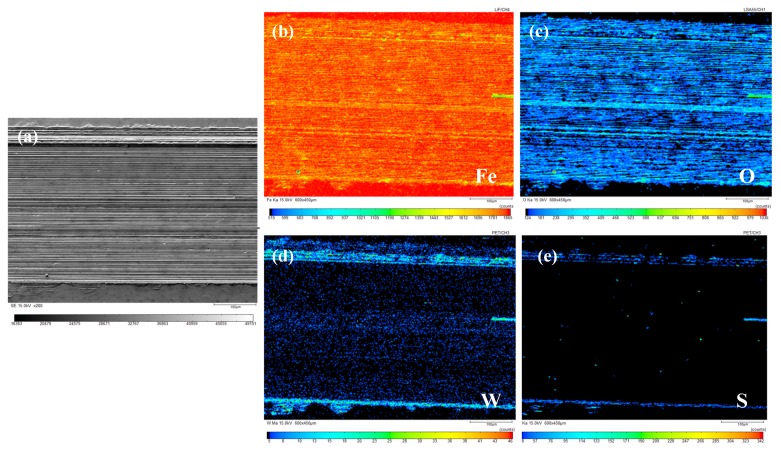
Element distribution diagram of wear scar of sample oil #2 (load: 120 N, sliding speed: 20 mm/s). (**a**) Scanning image; (**b**) Element Fe; (**c**) Element O; (**d**) Element W; (**e**) Element S.

**Figure 13 materials-13-01522-f013:**
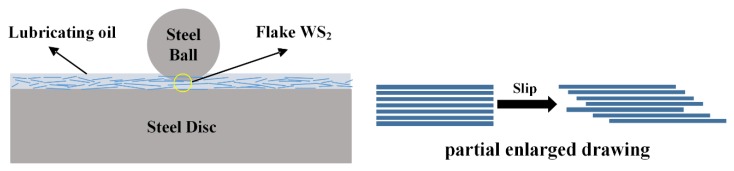
Lubrication mechanism of flaky WS_2_ as lubricating oil additive (steel ball and disk). (**Left**: overall schematic) (**Right**: schematic details).

**Figure 14 materials-13-01522-f014:**
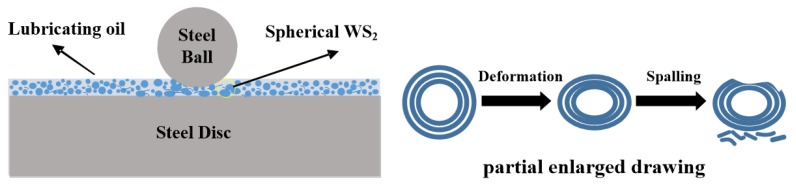
Lubrication mechanism of spherical WS_2_ as lubricating oil additive. (**Left**: overall schematic) (**Right**: schematic details).

**Table 1 materials-13-01522-t001:** Specific setting values of sliding velocity and constant load.

Parameter	Specific Set Value					
Sliding Velocity (mm/s)	10	15	20	25	30	35
Permanent Load (N)	20	40	60	80	100	120

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
