# Peer review of "Study on Tribological Properties and Mechanisms of Different Morphology WS2 as Lubricant Additives"

_materials, 2020, doi:10.3390/ma13071522_

Round 1
Reviewer 1 Report
Dear authors, i will consider this article for publication after ammending the comments. You can see the reviewed article attached.
Regards.

Author Response
We revised the words and sentences in the manuscript according to the comments and responded to the questions below.
- You also have to detail how did you calculate the wear rate. Mentioning the formula and any reference. For example: J. Ureña, E. Tabares, S. Tsipas, A. Jiménez-Morales, E. Gordo, “Dry sliding wear behaviour of β-type Ti-Nb and Ti-Mo surfaces designed by diffusion processes for biomedical applications”, J. Mech. Behav. Biomed. Mater., vol. 91, pp. 335-344, 2019.
Response: The wear losses of the material V can be obtained after the sliding tests were completed. Then according to classical wear equation, the wear rate W can be calculated by the following equation:
W=V/(S×L)
Where: V is the wear volume of the material; S is the sliding distance and L is the normal load applied
- What is ‘sommerfield’? What is ‘stribeck curve’?
Response: In general, Stribeck curve is used to reflect the transformation process of lubrication state and the relationship between friction factors and dimensionless characteristic number. The dimensionless characteristic number is also called the somemerfield number, which is defined as:
Where: is the dynamic viscosity of the lubricating oil; U is the sliding speed; p is the load pressure.
According to this curve, lubrication state can be divided into three main types: boundary lubrication, mixed lubrication and fluid lubrication.
Reviewer 2 Report
Line 146: Please correct the units " (20n, 60N, 100N and 120N)"
Authors should refer and include a recent tribological study on MWCNT-ionic liquid in introduction section:
https://books.google.com/books/about/Optimization_of_Wet_Friction_Systems_Bas.html?id=mzj4twEACAAJ&hl=en&output=html_text
Line 169 " main wear forms of wear 169 scars are plastic deformation, furrow cutting and adhesive wear. " please cite literature reference.
Line 190-191: Authors shall justify why the chemical reaction will occur and result in formation of Fe2O3 and FeO, not just simply stating these would occurs or are occurring. Else cite literature.
Line 193: Authors shall report the oxide ratios along with EDS/EDX spectras for clear distinction of WO3 vs WS2.
Line 212: To show the improvement a sample without WS2 must be included in the study. Currently missing?
Author Response
1.Authors should refer and include a recent tribological study on MWCNT-ionic liquid in introduction section:
Response: We appreciate your careful review on our manuscript. Since this paper does not involve any experiments related to ionic liquids, there is no reference to the latest tribological research of ionic liquids in the ‘Introduction’ section.
- Line 169 " main wear forms of wear 169 scars are plastic deformation, furrow cutting and adhesive wear. " please cite literature reference.
Response: References have been added as follows. This book provides us with a lot of basic theoretical knowledge about tribology and helps us to better understand the wear patterns of surface wear marks.
[28]Bhushan B . Introduction to Tribology, Second Edition[M]. 2013.
This literature introduces a method of measuring surface roughness curve area and volume by using image processing technology, which helps us to better understand the surface deformation mechanism and the wear problem of surface material.
[29]Nurul, Farhana, Mohd, et al. A technique to measure surface asperities plastic deformation and wear in rolling contact[J]. Wear, 2016.
This literature reviews the research progress of abrasive wear mechanism, analyzes the principle of detection method, introduces abrasive wear, fatigue wear and adhesive wear in detail, and provides the formula for calculating wear rate. The analysis of the literature plays a very important reference role.
[30]Mechanical wear debris feature,detection,and diagnosis:A review[J]. Chinese Journal of Aeronautics, 2018, v.31;No.146(05):5-20.
- Line 190-191: Authors shall justify why the chemical reaction will occur and result in formation of Fe2O3 and FeO, not just simply stating these would occurs or are occurring. Else cite literature.
Response: References have been added as follows.
The following two literatures both studied the tribological properties of WS2, analyzed the wear surface, and found that iron oxides were generated by chemical reactions during the sliding process, which provided a reference and theoretical basis for the wear mechanism analysis.
[31]Jiang Z , Zhang Y , Yang G , et al. Tribological Properties of Oleylamine-Modified Ultrathin WS2 Nanosheets as the Additive in Polyalpha Olefin Over a Wide Temperature Range[J]. Tribology Letters, 2016, 61(3):24.
[32]Zhengquan Jiang, Yang Guangbin, Zhang Yujuan, et al. Facile method preparation of oil-soluble tungsten disulfide nanosheets and their tribological properties over a wide temperature range[J]. Tribology International, 2019, 135287-295.
This literature explains the excellent tribological behavior of MoS2/WS2 quantum dots, which is due to the formation of a boundary lubrication film. This boundary lubrication film can be generated by the tribological reaction between MoS2/WS2 and iron atoms/iron oxide species, providing a reference for the wear mechanism.
[33]Wu X , Gong K , Zhao G , et al. MoS\r, 2\r, /WS\r, 2\r, Quantum Dots as High-Performance Lubricant Additive in Polyalkylene Glycol for Steel/Steel Contact at Elevated Temperature[J]. Advanced Materials Interfaces, 2018, 5(1):1700859.
The XPS spectra of MoS2/Gr nanocomposites were analyzed in this paper, found a friction film composed of MoS2, FeO/Fe3O4, FeSO4 / Fe2 (SO4) 3 and compound containing the C - O bonding on the lubricated metal surface, this literature provides reference for the analysis of friction mechanism.
[34]Wang, Xiaobo, Gong, et al. Nanosized MoS2 deposited on graphene as lubricant additive in polyalkylene glycol for steel/steel contact at elevated temperature[J]. Tribology International, 2017.
- Line 193: Authors shall report the oxide ratios along with EDS/EDX spectras for clear distinction of WO3 vs WS2.
Response: Thanks for your careful reading and insightful suggestions. The purpose of the element analysis of the worn surface is to confirm the residual elements on the surface. We did not consider the oxidation ratio to further distinguish WO3 and WS2, which is our omission.
- Line 212: To show the improvement a sample without WS2 must be included in the study. Currently missing?
Response: We appreciate your careful review on our manuscript. In the early stage, we have done research on the tribological properties of WS2, and made tribological tests of base oil and different added amounts of WS2. The research shows that adding WS2 can greatly improve the tribological properties of lubricants. [Na, Wu Ningning, Hu Gongbo, et al. Tribological properties of lubricating oil with micro/nano-scale WS2 particles[J]. Journal of Experimental Nanoscience, 2018.]
Based on the previous research, this paper focuses on the tribological properties of different forms of WS2 as lubricant additives under different loads, so a sample without WS2 was not added for comparison.
Reviewer 3 Report
The manuscript, by Hu et al., presents experimental studies of the influence of various morphology of the WS2 as lubricant additives on on tribological properties in different load and speed. The works were properly organised and conducted and the results well analysed to support conclusions. In my opinion the manuscript can be accepted in the present form.
Author Response
We appreciate your careful review of our manuscript, and thank you for your affirmation of our article.
Reviewer 4 Report
The article is well structured, the methodology well defined and the results obtained are well analyzed. It can be published in the current form.
Author Response

(The authors gave the same response as above.)

Reviewer 5 Report
The work is interesting especially for tribology society.
Some comments that can help to improve this work
The abstract should be reformulated…it is difficult to understand their message. For example first phrase is developed over 5 row that make extremely difficult to follow it. I underrated that your research refeer to WS2 but which material pairs were used
The state of art is very poor ..only 16 references are not enough. There are plenty materials tested as pair using ws2 an need be stressed here
The novelty of this work is not stressed well.
Did you bought this ws2 or manufactured in house ?
Why were used Steel balls ? and not other pair.
Why was selected this methodology ?
You suggested controlled humidity 40-50%, how did you control it ?
It is not clear that the friction coefficient was reduced because of ws2 or cause of load. It is obvious that the friction to be lower at higher load
For figure 5 you should use the same scale
The font size of Figure 6 and 7 is small please increase it
In Figure 8 for 20 mm/s at lower load you have the highest wear …little bit strength …normally I expect for bigger load to have higher wear. Besides there is not big difference between the two speed 15 and 20 do not make much difference
The results and discussion should be endorse by state of art observation
Author Response
- The abstract should be reformulated…it is difficult to understand their message. For example first phrase is developed over 5 row that make extremely difficult to follow it. I underrated that your research refeer to WS2 but which material pairs were used.
Response: Thanks for your careful reading and insightful suggestions. The abstract has been revised according to the comments.
- The state of art is very poor ..only 16 references are not enough. There are plenty materials tested as pair using ws2 an need be stressed here.
Response: We appreciate your careful review on our manuscript. References have been added to 35.
This literature describes various characteristics of WS2. WS2 is non-toxic, inert, non-magnetic and has high resistance to oxidation and thermal degradation.
[1]Olivas A , Villalpando I , S. Sepúlveda, et al. Synthesis and magnetic characterization of nanostructures N/WS 2, where N = Ni, Co and Fe[J]. Materials Letters, 2007, 61(21):4336-4339.
The tribological influence of WS2 as a solid lubricant on composites was studied, and it was found that WS2 can improve the tribological properties of Fe C Ni composites.
[2]Avi Gupta, Mohan Sanjay, Anand Ankush, et al. Tribological behaviour of Fe–C–Ni self-lubricating composites with WS2 solid lubricant[J]. Materials Research Express, 2019, 6(12): 126507.
The following two literatures mainly introduce the layered structure of WS2, which gives us a deeper understanding of the structure of WS2.
[3]Barrera, Diego, Lee, et al. Solution synthesis of few-layer 2H MX2 (M = Mo, W; X = S, Se)[J]. Journal of Materials Chemistry C Materials for Optical & Electronic Devices, 2017.
[4] José I. Martínez, Laikhtman A , Moon H R , et al. Modelling of adsorption and intercalation of hydrogen on/into tungsten disulphide multilayers and multiwall nanotubes[J]. Physical Chemistry Chemical Physics, 2018, 20.
This literature investigates the effect of dispersion of surface-modified WS2 nanoparticles on the tribological performance and physicochemical properties of motorbike lubricant. It was found that lubricant dispersed with WS2 nanoparticles gave good performance characterized by the reduction in both engine wear and fuel consumption.
[6]V Srinivas, Thakur R-N, Jain A-K, et al. Physicochemical properties and tribological performance of motorbike lubricant dispersed with surface-modified WS2 nanoparticles[J]. Proceedings of the Institution of Mechanical Engineers, Part J: Journal of Engineering Tribology, 2019, 233(9): 1379-1388.
This literature describes the preparation of 5nm ultra-thin ws2 nanosheet and tribological tests, and the result shows that ws2 can improve the wear resistance and wear reduction performance of the base oil.
[10]Zhang X , Xu H , Wang J , et al. Synthesis of Ultrathin WS2 Nanosheets and Their Tribological Properties as Lubricant Additives[J]. Nanoscale Research Letters, 2016, 11(1):442.
In this literature, the tribological behavior and formation of tribo-film of copper/tungsten disulfide (WS2) composites featuring 0 30% WS2 volume fractions, prepared using spark plasma sintering were investigated. Results indicated that WS2 as addition into the copper matrix could effectively reduce the coefficient of friction (COF) of Cu/WS2 composites.
[11]Effect of Adding Tungsten Disulfide to a Copper Matrix on the Formation of Tribo-Film and on the Tribological Behavior of Copper/Tungsten Disulfide Composites[J]. Tribology Letters, 2019, 67(3):1-13.
In this literature, 0.02-0.04 wt% WS2/GP was added into the base oil, and the results showed that WS2/GP dispersed oil showed significant lubrication performance
[12]Zheng D , Wu Y P , Li Z Y , et al. Tribological properties of WS 2 /graphene nanocompositees as lubricating oil additives[J]. Rsc Advances, 2017, 7(23):14060-14068.
(The above two literatures show that ws2 can not only be used as a lubricant additive to improve the lubrication performance, but also work together with other materials to improve the anti-wear and anti-wear performance.)
The tribological properties of lubricants containing WS2 nanoparticles were studied. It is worth mentioning that the boundary lubrication test method in this literature is similar to that in this literature.
[13]Raina A , Anand A . Effect of nanodiamond on friction and wear behavior of metal dichalcogenides in synthetic oil[J]. Applied Nanoscience, 2018(2):1-11.
This literature also makes use of UMT to study the tribological properties of WS2, and the research results are of great reference significance for this experiment
[14]X Zhang, Wang J, Xu H, et al. Preparation and Tribological Properties of WS2 Hexagonal Nanoplates and Nanoflowers[J]. Nanomaterials (Basel), 2019, 9(6).
In this literature, the mechanical properties and tribological behavior of tungsten disulfide and copper powder composites with different particle sizes were studied. The results indicate that the bending strength and tribological behavior of Cu−WS2 composites are greatly affected by the size of WS2 particles.
[17]Jin ZHOU, MA Chao, KANG Xiao, et al. Effect of WS 2 particle size on mechanical properties and tribological behaviors of Cu-WS 2 composites sintered by SPS[J]. Transactions of Nonferrous Metals Society of China, 2018, 28(6): 1176-1185.
[18]M, Zalaznik, M, et al. Effect of the type, size and concentration of solid lubricants on the tribological properties of the polymer PEEK[J]. Wear, 2016.
The tribological properties of solid lubricants of different sizes were investigated. And the above two literatures have shown that particles of different sizes have a great influence on the lubrication performance, which provides a theoretical basis for the tribological performance research of WS2 particles of different shapes.
- The novelty of this work is not stressed well. Did you bought this ws2 or manufactured in house ?
Response: The two kinds of WS2 used in the experiment was purchased from Sigma-Aldrich Corporation. Then, the WS2 was characterized (SEM, XRD and TEM) to determine its surface morphology, particle size and chemical composition.
- Why were used Steel balls ? and not other pair.
Response: The friction material GCr15 steel selected in this experiment is a common high carbon chromium bearing steel, which has good wear resistance, uniform structure, high hardness and other properties. The anti-friction and anti-wear behavior of lubricating oil depends to a large extent on the material of the mating pair. Some experiments have shown that the wear surface morphology of steel plate is different under different mating pair. The wear surface of steel plate is better with GCr15 steel discs / GCr15 steel balls, which is convenient to observe its wear morphology.
- Why was selected this methodology ?
Response: Firstly, a UMT-2 friction and wear tester was selected for this experiment. The UMT-2 friction and wear tester runs smoothly and has low mechanical vibration. It is the most versatile nano- / micron-level friction and wear tester and can well meet the requirements of this test. Secondly, the ball-disk friction can better observe the wear of the friction surface. Finally, drip lubrication was used in this test. This method is suitable for low-speed and light-load working conditions with low temperature.
- You suggested controlled humidity 40-50%, how did you control it ?
Response: There are two operation methods to adjust the air moisture content, namely, humidification and dehumidification. If humidification is needed, humidifier can be used, and if dehumidification is needed, wheel dehumidifier can be used. The humidity range is specified in this article to indicate that the experiment is completed in a period of time when the humidity is stable, and the experiment is completed in a short period of time.
- It is not clear that the friction coefficient was reduced because of ws2 or cause of load. It is obvious that the friction to be lower at higher load.
Response: Earlier, we studied the tribological properties of two types of WS2, and studied the effect of the amount of WS2 added on the tribological properties of the base oil. The results show that the addition of WS2 can significantly improve the anti-friction and anti-wear properties of lubricants. [Na, Wu Ningning, Hu Gongbo, et al. Tribological properties of lubricating oil with micro/nano-scale WS2 particles[J]. Journal of Experimental Nanoscience, 2018.]
In addition, when the load is small, the thickness of the oil film is thick, and the contact surface of the friction pair is completely separated by the oil film. The friction force at this time comes from the viscous shear of the lubricating oil. Therefore, when the load is small, the friction coefficient is large, which is most likely caused by the viscous shear force of the lubricating oil.
- For figure 5 you should use the same scale. The font size of Figure 6 and 7 is small please increase it.
Response: :We appreciate your careful review on our manuscript. Figure 5, Figure 6, and Figure 7 have been revised as recommended. The coordinates of Figure 5 have been unified. Figures 6 and 7 are enlarged.
- In Figure 8 for 20 mm/s at lower load you have the highest wear …little bit strength …normally I expect for bigger load to have higher wear. Besides there is not big difference between the two speed 15 and 20 do not make much difference.
Response: Fig. 8 shows the wear rate of sample oil containing flake WS2 at 15mm / s and sample oil containing spherical WS2 at 20mm / s. The wear rate of the two samples is similar under different loads, but their values are still different. Obviously, the wear resistance of spherical WS2 is better. Although the wear rate of low load is higher at 20 mm / s, the trend of wear rate is similar to that of 15 mm / s with the increase of load. The high value may be caused by experimental error.
- The results and discussion should be endorse by state of art observation.
Response: Thanks for your careful reading and insightful suggestions. The results and discussion in this paper have improved.
Reviewer 6 Report
The reviewed paper ”Study on tribological properties and mechanism of different morphology WS2 as lubricant additives” seems to be interesting and high potential. In my opinion authors should consider the significantly expanding of introduction (e.g. the role of MoS2 as a lubricant was treated very generally) and experimental part (no details about the apparatus used and its measurement settings). The paper contains many errors/mistakes that must be eliminated. Few of them are listed below:
- Page 1, Line: 31: Should be ”from -273 °C to ~ +425 °C rather than ”-273 ~ 425 °C”.
- Page 1, Line: 32: Should be ”Aldana et al.” rather than ”Aldana P U et al.”.
- Page 1, Line: 33: Acronyms PAO and ZDDP have not been defined.
- Page 1, Line: 41: Between value and unit should be used a separator – “90 nm” rather than ”90nm”. This major remark applies to the entire manuscript – please carefully check and correct.
- Page 1, Introduction: Please improve the introduction by adding a more recent and relevant literature.
- Page 2, Line: 61/62: Should be ”Scanning Electron Microscopy” rather than ”scanning electron microscopy”.
- Page 2, Line: 61/62: There is no detailed information (type, producer, parameters of imaging/measurements) about the used apparatus (XRD, SEM, TEM).
- Page 2, Line: 64: The term ”Figure” should be used in the text, whereas ”Fig.” in brackets. Please carefully check and correct this mistake in the entire manuscript.
- Page 2, Line: 72: XRD image? XRD technique is used for imaging? There are no units on the axes.
- Page 3, Line: 75: The SEM-micrographs shown seems to be very random. Why they were not presented in a thoughtful order (from the smallest to the largest magnification). Is Fig. 2a a fragment of Fig. 2b?
- Page 3, Line: 78: Magnifications for both TEM-micrographs are not given.
- Page 3, Line: 81: What is the producer of UMT-2 tribometer?
- Page 3, Line: 83: Please eliminate the conjunction hanging at the end of the line – move "A" to the next line.
- Page 3, Line: 86: Please use the correct nomenclature in relation to the surface roughness parameters. I strongly recommend to review the ISO 4287(1997) standard.
- Page 3, Line: 88: Should be ”Table 1” rather than ”table 1”.
- Page 3, Line: 95/96: What is the producer of SV-3000 Ultra-Depth Three-Dimensional Microscope?
- Page 4, Line: 97: There is no detailed information (type, producer, parameters of imaging/measurements) about the electron-probe micro-analyzer (EPMA).
- Page 4, Figure 4: Friction coefficient has any unit?
- Page 5, Figure 5: Is it about the Sommerfeld number (axis x)?
- Page 5, Line: 146: Should be ”20 N” rather than ”20n”.
- Page 5, Line: 153-154: Should be ”×10-4mm3·N-1m-1” rather than ”×10-4mm3·N-1m-1”.
- Page 6, Line: 159/162: The term "three-dimensional morphologies" is a bit confusing.
- Page 6, Line: 167: Should be ”SEM-micrographs” rather than ”SEM diagrams”.
- Page 7, Figure 9-10: No detail description regarding (a)-(d).
- The authors use many symbols and acronyms. I suggest consider inserting their explanation in the Nomenclature at the end of the work (after the Conflicts of Interest).
- Please carefully check English language and grammar.
Author Response
- Page 1, Line: 31: Should be ”from -273 °C to ~ +425 °C rather than ”-273 ~ 425 °C”.
Response: Thanks for your careful reading and insightful suggestions. It has been corrected.
- Page 1, Line: 32: Should be ”Aldana et al.” rather than ”Aldana P U et al.”.
Response: We appreciate your careful review of our manuscript. It has been corrected.
- Page 1, Line: 33: Acronyms PAO and ZDDP have not been defined.
Response: It has been added to our manuscript.
Poly Alpha Olefin (PAO) is a kind of synthetic base oil. It is made from alpha olefin through the polymerization of ethylene, and then further polymerized and hydrogenated. It is the most commonly used synthetic lubricant base oil and has the most widely used.
Zinc dialkyl dithiophosphate (ZDDP), as a common anti-wear additive for lubricating oils, has been used for more than 70 years and is still the critical component of almost all modern gasoline and diesel engine lubricants.
- Page 1, Line: 41: Between value and unit should be used a separator – “90 nm” rather than ”90nm”. This major remark applies to the entire manuscript – please carefully check and correct.
Response: Thank you for your reminding, we have carefully examined the full text and corrected it one by one.
5.Page 1, Introduction: Please improve the introduction by adding a more recent and relevant literature.
Response: Some of the latest references have been added to the introduction in this paper.
- Page 2, Line: 61/62: Should be ”Scanning Electron Microscopy” rather than ”scanning electron microscopy”.
Response: Thanks for your careful reading, it has been corrected.
- Page 2, Line: 61/62: There is no detailed information (type, producer, parameters of imaging/measurements) about the used apparatus (XRD, SEM, TEM).
Response: We have all the details of these devices, but we haven't written in the article.
X-Ray Diffraction (XRD)
Type: D8 Advance
Producer: Bruker company in Germany
Parameters of imaging/measurements: It is a powerful experimental method to determine the crystal structure and its changing law, and also a common analytical research method to explore the internal relationship between phase composition and macroscopic properties of the microstructure of materials.
Using Cu target, Ka radiation, scanning speed of 0.2sec/step, sampling interval of 0.01945°(step), scanning Angle (2q) range of 10°-70°, operating voltage of 40kV, operating current of 30mA.
Scanning Electron Microscope (SEM)
Type: Quanta 250
Producer: FEI company in the United States
Parameters of imaging/measurements: The device is an indispensable tool for exploring the microscopic world and characterizing surface structure and composition, commonly used in micro/nano materials and the wear surface traces of the composition analysis and microstructure observation.
The equipment can conduct direct analysis and characterization of conductor/semiconductor/insulator in high/low/ultra-low vacuum environment without spraying conductive layer, and the amplification factor is 6 times ~ 1 million times.
Transmission Electron Microscopy (TEM)
Type: Tecnai G2 F20
Producer: FEI company in the United States
Parameters of imaging/measurements: The equipment is mainly used for qualitative and quantitative analysis of microchemical composition of mineral rocks and materials (ceramics, alloys, semiconductor materials, etc.), chemical composition surface and chemical composition phase analysis in micro-area, and micro area morphology observation, etc.
The point resolution is 0.24nm, the line resolution is 0.102nm, the information resolution is 0.14nm, and the magnification is 25-1.03 million times.
- Page 2, Line: 64: The term ”Figure” should be used in the text, whereas ”Fig.” in brackets. Please carefully check and correct this mistake in the entire manuscript.
Response: We appreciate your careful review of our manuscript. We have carefully examined the full text and corrected it one by one.
- Page 2, Line: 72: XRD image? XRD technique is used for imaging? There are no units on the axes.
Response: We drew the image according to the experimental data. The abscissa should be the angle, which is the angle between the incident light and the reflected light. The ordinate should be a count strength value, which can be any unit.
- Page 3, Line: 75: The SEM-micrographs shown seems to be very random. Why they were not presented in a thoughtful order (from the smallest to the largest magnification). Is Fig. 2a a fragment of Fig. 2b?
Response: Fig. 2a is not a fragment of Fig. 2b. Fig. 2a and Fig. 2b are two magnifications of the same batch of samples. They were chosen because they are photographed clearly.
- Page 3, Line: 78: Magnifications for both TEM-micrographs are not given.
Response: In TEM characterization, only the scale is given, and no magnification is given.
- Page 3, Line: 81: What is the producer of UMT-2 tribometer?
Response: UMT-2 tribometer is produced by CETR corporation in America.
- Page 3, Line: 83: Please eliminate the conjunction hanging at the end of the line – move "A" to the next line.
Response: We appreciate your careful review of our manuscript. It has been corrected.
- Page 3, Line: 86: Please use the correct nomenclature in relation to the surface roughness parameters. I strongly recommend to review the ISO 4287(1997) standard.
Response: Thanks for your insightful comments, we have revised it based on the suggestions.
- Page 3, Line: 88: Should be ”Table 1” rather than ”table 1”.
Response: It has been corrected.
- Page 3, Line: 95/96: What is the producer of SV-3000 Ultra-Depth Three-Dimensional Microscope?
Response: SV-3000 Ultra-Depth Three-Dimensional Microscope is produced by Kanagawa Sanfeng company in Japan.
- Page 4, Line: 97: There is no detailed information (type, producer, parameters of imaging/measurements) about the electron-probe micro-analyzer (EPMA).
Response: Electron Probe Microanalyzer (EPMA)
Type: 8050G
Producer: Shimadzu company of Japan
parameters of imaging/measurements: The equipment is mainly used for qualitative and quantitative analysis of microchemical composition of mineral rocks and materials (ceramics, alloys, semiconductor materials, etc.), chemical composition surface and chemical composition phase analysis in micro-area, and micro area morphology observation, etc.
The scanning range of elements is 4Be ~ 92U, the channel of spectrometer is 4 channels, the X ray extraction Angle is 52.5°, the magnification is 40 ~ 400000 times.
- Page 4, Figure 4: Friction coefficient has any unit?
Response: The friction coefficient is a ratio and has no units.
- Page 5, Figure 5: Is it about the Sommerfeld number (axis x)?
Response: Yes, it is. In general, stribeck curve is used to reflect the transformation process of lubrication state and the relationship between friction factors and dimensionless characteristic number. The dimensionless characteristic number is also called the somemerfield number, which is defined as:
Where: is the dynamic viscosity of the lubricating oil; U is the sliding speed; p is the load pressure.
- Page 5, Line: 146: Should be ”20 N” rather than ”20n”.
Response: It has been corrected.
- Page 5, Line: 153-154: Should be ”×10-4mm3·N-1m-1” rather than ”×10-4mm3·N-1m-1”.
Response: It has been corrected.
- Page 6, Line: 159/162: The term "three-dimensional morphologies" is a bit confusing.
Response: The three-dimensional topography measuring instrument is used to measure the topography and the surface of the microstructure. “three-dimensional morphologies” should be “three-dimensional topography”.
- Page 6, Line: 167: Should be ”SEM-micrographs” rather than ”SEM diagrams”.
Response: It has been corrected.
- Page 7, Figure 9-10: No detail description regarding (a)-(d).
Response: In Figure 9-10, the upper right corner of (a) - (d) is marked with corresponding experimental load and sliding speed.
- The authors use many symbols and acronyms. I suggest consider inserting their explanation in the Nomenclature at the end of the work (after the Conflicts of Interest).
Response: Thanks for your careful reading and insightful suggestions. The symbols and abbreviations of the full text have been carefully examined and explained.
- Please carefully check English language and grammar.
Response: We checked the language and grammar of the whole paper and have tried our best to improve the manuscript and made some changes in the manuscript. These changes will not influence the content and framework of the paper. And here we did not list the changes but marked in highlight in revised paper.
Round 2
Reviewer 2 Report
- Line 169 " main wear forms of wear 169 scars are plastic deformation, furrow cutting and adhesive wear. " please cite literature reference.
Response: References have been added as follows. This book provides us with a lot of basic theoretical knowledge about tribology and helps us to better understand the wear patterns of surface wear marks.
[28]Bhushan B . Introduction to Tribology, Second Edition[M]. 2013.
This literature introduces a method of measuring surface roughness curve area and volume by using image processing technology, which helps us to better understand the surface deformation mechanism and the wear problem of surface material.
[29]Nurul, Farhana, Mohd, et al. A technique to measure surface asperities plastic deformation and wear in rolling contact[J]. Wear, 2016.
This literature reviews the research progress of abrasive wear mechanism, analyzes the principle of detection method, introduces abrasive wear, fatigue wear and adhesive wear in detail, and provides the formula for calculating wear rate. The analysis of the literature plays a very important reference role.
[30]Mechanical wear debris feature,detection,and diagnosis:A review[J]. Chinese Journal of Aeronautics, 2018, v.31;No.146(05):5-20.
Round 2 Comment: A book reference shall also include page numbers.
- Line 190-191: Authors shall justify why the chemical reaction will occur and result in formation of Fe2O3 and FeO, not just simply stating these would occurs or are occurring. Else cite literature.
Response: References have been added as follows.
The following two literatures both studied the tribological properties of WS2, analyzed the wear surface, and found that iron oxides were generated by chemical reactions during the sliding process, which provided a reference and theoretical basis for the wear mechanism analysis.
[31]Jiang Z , Zhang Y , Yang G , et al. Tribological Properties of Oleylamine-Modified Ultrathin WS2 Nanosheets as the Additive in Polyalpha Olefin Over a Wide Temperature Range[J]. Tribology Letters, 2016, 61(3):24.
[32]Zhengquan Jiang, Yang Guangbin, Zhang Yujuan, et al. Facile method preparation of oil-soluble tungsten disulfide nanosheets and their tribological properties over a wide temperature range[J]. Tribology International, 2019, 135287-295.
This literature explains the excellent tribological behavior of MoS2/WS2 quantum dots, which is due to the formation of a boundary lubrication film. This boundary lubrication film can be generated by the tribological reaction between MoS2/WS2 and iron atoms/iron oxide species, providing a reference for the wear mechanism.
[33]Wu X , Gong K , Zhao G , et al. MoS\r, 2\r, /WS\r, 2\r, Quantum Dots as High-Performance Lubricant Additive in Polyalkylene Glycol for Steel/Steel Contact at Elevated Temperature[J]. Advanced Materials Interfaces, 2018, 5(1):1700859.
The XPS spectra of MoS2/Gr nanocomposites were analyzed in this paper, found a friction film composed of MoS2, FeO/Fe3O4, FeSO4 / Fe2 (SO4) 3 and compound containing the C - O bonding on the lubricated metal surface, this literature provides reference for the analysis of friction mechanism.
[34]Wang, Xiaobo, Gong, et al. Nanosized MoS2 deposited on graphene as lubricant additive in polyalkylene glycol for steel/steel contact at elevated temperature[J]. Tribology International, 2017.
Round 2 Comment: Best to add a brief summary of justification stated here in manuscript as well since this is basis for further discussion in paper.
- Line 193: Authors shall report the oxide ratios along with EDS/EDX spectras for clear distinction of WO3 vs WS2.
Response: Thanks for your careful reading and insightful suggestions. The purpose of the element analysis of the worn surface is to confirm the residual elements on the surface. We did not consider the oxidation ratio to further distinguish WO3 and WS2, which is our omission.
Round 2 Comment: Fig 11(d) shows W all throughout while Fig 11-e shows minimal (extremely low detection) of S element which can appear in EDS mapping in error as well. Hence it is must to report the OXIDE ratios for clear distinction between WO3 and WS2 using oxide ratio quanitification.
- Line 212: To show the improvement a sample without WS2 must be included in the study. Currently missing?
Response: We appreciate your careful review on our manuscript. In the early stage, we have done research on the tribological properties of WS2, and made tribological tests of base oil and different added amounts of WS2. The research shows that adding WS2 can greatly improve the tribological properties of lubricants. [Na, Wu Ningning, Hu Gongbo, et al. Tribological properties of lubricating oil with micro/nano-scale WS2 particles[J]. Journal of Experimental Nanoscience, 2018.]
Based on the previous research, this paper focuses on the tribological properties of different forms of WS2 as lubricant additives under different loads, so a sample without WS2 was not added for comparison.
Round 2: It is basic customary to include a CONTROL sample in a engineering study.
Author Response
- Round 2 Comment: A book reference shall also include page numbers.
Respond: As reviewer suggested that we added page numbers to the book literature.
[28]Bhushan B . Introduction to Tribology, Second Edition[M]. 2013.P315-396.
- Round 2 Comment: Best to add a brief summary of justification stated here in manuscript as well since this is basis for further discussion in paper.
Respond: Thank you for your valuable suggestions. We have added a brief summary of justification to support our experimental analysis and discussion in the manuscript and highlighted it.
The details are as follows: According to literature [31-35], it was found that oxidation products were formed on the friction surface through the X-ray Photoelectron Spectroscopy (XPS) analysis of the friction surface after the test. After further study on the products formed, it was found that the Fe 2p peak corresponds to Fe2O3 or FeO, and the worn surface appeared FeO, Fe2O3 film. The XPS data indicate that a frictional chemical reaction occurs on the surface of the wear steel during the friction process, thus forming the boundary lubrication film composed of WO3, Fe2O3 and FeO.
[35]Fan X, Wang L, Li W, et al. Improving Tribological Properties of Multialkylated Cyclopentanes under Simulated Space Environment: Two Feasible Approaches[J]. Acs Applied Materials & Interfaces, 2015, 7(26):14359-14368.
- Round 2 Comment: Fig 11(d) shows W all throughout while Fig 11-e shows minimal (extremely low detection) of S element which can appear in EDS mapping in error as well. Hence it is must to report the OXIDE ratios for clear distinction between WO3 and WS2 using oxide ratio quanitification.
Respond: From Figures 11 (d) and (e), it can be seen that the content of element W is significantly more than that of element S. These scattered elements S are most likely deposited WS2, which were confirmed in another article of ours. [Wu N, Hu N, Wu J, et al. Tribology Properties of Synthesized Multiscale Lamellar WS2 and Their Synergistic Effect with Anti-Wear Agent ZDDP[J]. Applied Sciences, 2020, 10(1): 115.] We have captured the residual WS2 through SEM, as shown in the figure below. Therefore, we have reasons to believe that the S element here is not an error, but a small amount of WS2 remaining on the worn surface.
- Round 2: It is basic customary to include a CONTROL sample in a engineering study.
Respond: In this paper, flake WS2 and spherical WS2 are the main comparison objects, which can be used as a control group for each other. On the premise that WS2 can improve the anti-friction and anti-wear performance of base oils, we have not set a control sample. However, in the future research, we will pay attention to this detail, and thank you for your valuable reminder.
Reviewer 5 Report
The authors improved the manuscript acordingly to reviewers comments therefore, this paper can be considered for publication.
Author Response
Thank you for your careful reading and meaningful comments on our manuscript. Your suggestions make our article more prominent.